# CDTrans: Cross-domain Transformer for Unsupervised Domain Adaptation

**Tongkun Xu**[12*]**, Weihua Chen**[1*]**, Pichao Wang**[1]**, Fan Wang**[1]**, Hao Li**[1]**, Rong Jin**[1]
[1]Alibaba Group, [2]Shandong University
`xutongkun1208@gmail.com, kugang.cwh@alibaba-inc.com`
`{pichao.wang,fan.w,lihao.lh,jinrong.jr}@alibaba-inc.com`

## Abstract

Unsupervised domain adaptation (UDA) aims to transfer knowledge learned from a labeled source domain to a different unlabeled target domain. Most existing UDA methods focus on learning domain-invariant feature representation, either from the domain level or category level, using convolution neural networks (CNNs)-based frameworks. One fundamental problem for the category level based UDA is the production of pseudo labels for samples in target domain, which are usually too noisy for accurate domain alignment, inevitably compromising the UDA performance. With the success of Transformer in various tasks, we find that the cross-attention in Transformer is robust to the noisy input pairs for better feature alignment, thus in this paper Transformer is adopted for the challenging UDA task. Specifically, to generate accurate input pairs, we design a two-way center-aware labeling algorithm to produce pseudo labels for target samples. Along with the pseudo labels, a weight-sharing triple-branch transformer framework is proposed to apply self-attention and cross-attention for source/target feature learning and source-target domain alignment, respectively. Such design explicitly enforces the framework to learn discriminative domain-specific and domain-invariant representations simultaneously. The proposed method is dubbed CDTrans (cross-domain transformer), and it provides one of the first attempts to solve UDA tasks with a pure transformer solution. Experiments show that our proposed method achieves the best performance on public UDA datasets, e.g. VisDA-2017 and DomainNet. Code and models are available at `https://github.com/CDTrans/CDTrans`.

## 1 Introduction

Deep neural network have achieved remarkable success in a wide range of application scenarios (Wang et al., 2022; Qian et al., 2021; Yiqi Jiang, 2022; Tan et al., 2019; Chen et al., 2021b; Jiang et al., 2021; Chen et al., 2017) but it still suffers poor generalization performance to other new domain because of the domain shift problem (Csurka, 2017; Zhao et al., 2020; Zhang et al., 2020; Oza et al., 2021). To handle this issue and avoid the expensive laborious annotations, lots of research efforts (Bousmalis et al., 2017; Kuroki et al., 2019; Wilson & Cook, 2020; VS et al., 2021) are devoted on Unsupervised Domain Adaptation (UDA). The UDA task aims to transfer knowledge learned from a labeled source domain to a different unlabeled target domain. In UDA, most approaches focus on aligning distributions of source and target domain and learning domain-invariant feature representations. One kind of such UDA methods are based on category-level alignment (Kang et al., 2019; Zhang et al., 2019; Jiang et al., 2020; Li et al., 2021b), which have achieved promising results on public UDA datasets using deep convolution neural networks (CNNs). The fundamental problems in category-level based alignment is the production of pseudo labels for samples in target domain to generate the input source-target pairs. However, the current CNNs-based methods are not robust to the generated noisy pseudo labels for accurate domain alignment (Morerio et al., 2020; Jiang et al., 2020).

With the success of Transformer in natural language processing (NLP) (Vaswani et al., 2017; Devlin et al., 2018) and vision tasks (Dosovitskiy et al., 2020; Han et al., 2020; He et al., 2021; Khan et al.,

---

*These authors contributed equally to this work.

2021), it is found that cross-attention in Transformer is good at aligning different distributions, even from different modalities *e.g.*, vision-to-vision (Li et al., 2021e), vision-to-text (Tsai et al., 2019; Hu & Singh, 2021) and text-to-speech (Li et al., 2019). And we find that it is robust to noise in pseudo labels to some extent. Hence, in this paper, we apply transformers to the UDA task to take advantage of its robustness to noise and super power for feature alignment to deal with the problems as described above in CNNs.

In our experiment, we conclude that even with noise in the labeling pair, the cross-attention can still work well in aligning two distributions, thanks to the attention mechanism. To obtain more accurate pseudo labels, we designed a two-way center-aware labeling algorithm for samples in the target domain. The pseudo labels are produced based on the cross-domain similarity matrix, and a center-aware matching is involved to weight the matrix and weaken noise into the tolerable range. With the help of pseudo labels, we design the cross-domain transformer (CDTrans) for UDA. It consists of three weight-sharing transformer branches, of which two branches are for source and target data respectively and the third one is the feature alignment branch, whose inputs are from source-target pairs. The self-attention is applied in the source/target transformer branches and cross-attention is involved in the feature alignment branch to conduct domain alignment. Such design explicitly enforces the framework to learn discriminative domain-specific and domain-invariant representations simultaneously. In summary, our contributions are three-fold:

- We propose a weight-sharing triple-branch transformer framework, namely, CDTrans, for accurate unsupervised domain adaptation, taking advantage of its robustness to noisy labeling data and great power for feature alignment.
- To produce pseudo labels with high quality, a two-way center-aware labeling method is proposed, and it boosts the final performance in the context of CDTrans.
- CDTrans achieves the best performance compared to state-of-the-arts with a large margin on VisDA-2017 (Peng et al., 2017) and DomainNet (Peng et al., 2019) datasets.

## 2 RELATED WORK

### 2.1 TRANSFORMER FOR VISION

Transformer is proposed in (Vaswani et al., 2017) to model sequential data in the field of NLP. Many works have shown its effectiveness for computer-vision tasks (Han et al., 2020; Khan et al., 2021; Li et al., 2021d; Han et al., 2021b; Yu et al., 2021; Li et al., 2021c; Yang et al., 2021; Qian et al., 2022). Pure Transformer based models are becoming more and more popular. For example, ViT (Dosovitskiy et al., 2020) is proposed recently by feeding transformer with sequences of image patches; Touvron et al. (Touvron et al., 2021) propose DeiT that introduces a distillation strategy for transformers to help with ViT training; many other ViT variants (Yuan et al., 2021a; Wang et al., 2021; Han et al., 2021a; Chen et al., 2021a; Ranftl et al., 2021; Liu et al., 2021) are proposed from then, which achieve promising performance compared with its counterpart CNNs for both image classification and downstream tasks, such as object detection (Liu et al., 2021), semantic segmentation (Yuan et al., 2021b) and object ReID (He et al., 2021). For multi-modal based networks, there are several works (Tsai et al., 2019; Li et al., 2021e; Hu & Singh, 2021) that apply cross-attention for multi-modal feature fusion, which demonstrates that attention mechanism is powerful at distilling noise and feature alignment. This paper adopts cross-attention in the context of pure transformers for UDA tasks.

### 2.2 UNSUPERVISED DOMAIN ADAPTATION

There are mainly two levels for UDA methods: domain-level (Tzeng et al., 2014; Long et al., 2015; Ghifary et al., 2016; Tzeng et al., 2017; Bousmalis et al., 2017; Hoffman et al., 2018) and category-level (Saito et al., 2018; Kang et al., 2019; Du et al., 2021; Li et al., 2021a). Domain-level UDA mitigates the distribution divergence between the source and target domain by pulling them into the same distribution at different scale levels. The commonly used divergence measures include Maximum Mean Discrepancy (MMD) (Gretton et al., 2006; Tzeng et al., 2014; Long et al., 2015) and Correlation Alignment (CORAL) (Sun et al., 2016; Sun & Saenko, 2016). Recently, some works (Saito et al., 2018; Du et al., 2021; Li et al., 2021a) focus on the fine-grained category-level

label distribution alignment through an adversarial manner between the feature extractor and two domain-specific classifiers. Unlike coarse-grained alignment at the domain scale, this approach aligns each category distribution between the source and target domain data by pushing the target samples to the distribution of source samples in each category. Obviously, the fine-grained alignment results in more accurate distribution alignment within the same label space. Although the adversarial approach achieves new improvements by fusing fine-grained alignment operations of source and target samples at the category level, it still does not solve the problem of noisy samples in the wrong category. Our method adopts Transformers for category-level UDA to solve the noise problem.

## 2.3 Pseudo Labeling

Pseudo labeling (Lee et al., 2013) is first introduced for semi-supervised learning and gains popularity in domain adaptation tasks. It learns to label unlabeled data using predicted probabilities and performs fine-tuning together with labeled data. In terms of using pseudo labeling for domain adaptation tasks, (Long et al., 2017; 2018) adopt pseudo labels to conduct conditional distribution alignment; (Zhang et al., 2018; Choi et al., 2019) use pseudo labels as a regularization for domain adaptation; Zou et al. (2018) designs a self-training framework by alternately solving pseudo labels; Caron et al. (2018) propose a deep self-supervised method by generating pseudo labels via $k$-means cluster to progressively train the model; Liang et al. (2020) develop a self-supervised pseudo labeling method to alleviate the effects of noisy pseudo labels. Based on Liang et al. (2020), in this work, we propose a two-way center-aware labeling algorithm to further filter the noisy pseudo pairs.

## 3 The Proposed Method

We first introduce the cross attention module and analyze its robustness to the noise in Section 3.1. Then the two-way center-aware labeling method is presented in Section 3.2. With the produced pseudo labels as inputs, our cross-domain transformer (CDTrans) is proposed in Section 3.3, consisting of three weight-sharing transformers.

### 3.1 The Cross Attention in Transformer

#### 3.1.1 Preliminary

Vision Transformer (ViT) (Dosovitskiy et al., 2020) has achieved comparable or even superior performance on computer vision tasks. One of the most important structures in ViT is the self-attention module (Vaswani et al., 2017). In ViT, an image $\boldsymbol{I} \in \mathbb{R}^{H \times W \times C}$ is reshaped into a sequence of flattened 2D patches $x \in \mathbb{R}^{N \times (P^2 \cdot C)}$, where $(H, W)$ is the resolution of the original image, $C$ is the number of channels, $(P, P)$ is the resolution of each image patch, and $N = HW/P^2$ is the resulting number of patches. For self-attention, the patches are first projected into three vectors, *i.e.* queries $\boldsymbol{Q} \in \mathbb{R}^{N \times d_k}$, keys $\boldsymbol{K} \in \mathbb{R}^{N \times d_k}$ and values $\boldsymbol{V} \in \mathbb{R}^{N \times d_v}$. $d_k$ and $d_v$ indicates their dimensions. The output is computed as a weighted sum of the values, where the weight assigned to each value is computed by a compatibility function of the query with the corresponding key. The $N$ patches serve as the inputs for the self-attention module, and the process can be formulated as below. The self-attention module aims to emphasize relationships among patches of the input image.

$$Attn_{self}(\boldsymbol{Q}, \boldsymbol{K}, \boldsymbol{V}) = softmax(\frac{\boldsymbol{Q}\boldsymbol{K}^T}{\sqrt{d_k}})\boldsymbol{V} \tag{1}$$

The cross-attention module is derived from the self-attention module. The difference is that the input of cross-attention is a pair of images, *i.e.* $\boldsymbol{I}_s$ and $\boldsymbol{I}_t$. Its query and key/value are from patches of $\boldsymbol{I}_s$ and $\boldsymbol{I}_t$ respectively. The cross-attention module can be calculated as follows:

$$Attn_{cross}(\boldsymbol{Q}_s, \boldsymbol{K}_t, \boldsymbol{V}_t) = softmax(\frac{\boldsymbol{Q}_s\boldsymbol{K}_t^T}{\sqrt{d_k}})\boldsymbol{V}_t \tag{2}$$

where $\boldsymbol{Q}_s \in \mathbb{R}^{M \times d_k}$ are queries from $M$ patches of image $\boldsymbol{I}_s$, and $\boldsymbol{K}_t \in \mathbb{R}^{N \times d_k}, \boldsymbol{V}_t \in \mathbb{R}^{N \times d_v}$ are keys and values from $N$ patches of image $\boldsymbol{I}_t$. The output of the cross-attention module holds the same length $M$ as the number of the queries. For each output, it is calculated by multiplying $\boldsymbol{V}_t$

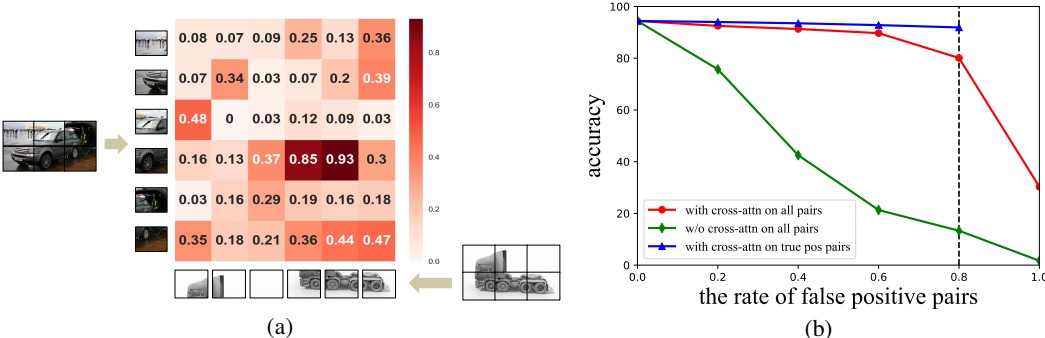

(a)                                    (b)

Figure 1: (a): The heatmap of the cross-attention weights for a false positive pair (Car vs. Truck). (b): The changes of UDA performance by the ratio of false positive pairs. The red/green curves represent the model with and without the cross-attention module. The blue curve means that only true positive pairs are involved in the cross-attention module.

with attention weights, which comes from the similarity between the corresponding query in $I_s$ and all the keys in $I_t$. As a result, among all patches in $I_t$, the patch that is more similar to the query of $I_s$ would hold a larger weight and contribute more to the output. In other words, the output of the cross-attention module manages to aggregate the two input images based on their similar patches.

So far, many researchers have utilized the cross-attention for feature fusion, especially in multi-modal tasks (Tsai et al., 2019; Li et al., 2019; Hu & Singh, 2021; Li et al., 2021e). In these works, the inputs of the cross-attention module are from two modalities, *e.g.* vision-to-text (Tsai et al., 2019; Hu & Singh, 2021), text-to-speech (Li et al., 2019) and vision-to-vision (Li et al., 2021e). They apply the cross-attention to aggregate and align the information from two modalities. Given its great power in feature alignment, we propose to use the cross attention module to solve the unsupervised domain adaptation problem.

### 3.1.2 ROBUSTNESS TO NOISE

As mentioned above, the input of the cross-attention module is a pair of images, which usually comes from two domains, and the cross-attention module aims to align these two images. If label noise exists, there would be false positive pairs in the training data. Images in the false positive pairs would have dissimilar appearance, and forcibly aligning their features would inevitably injure the training and compromise the performance. We assume that the dissimilar patches in false positive pairs are more harmful to the performance than the similar patches. In the cross-attention module, two images are aligned based on their patch similarity. As shown in Fig. 1a, the cross-attention module would assign a low weight to the dissimilar patches in false positive pairs. Thus it weakens the negative effects of the dissimilar patches on the final performance to some extent. [1]

To further analyze this issue, an experiment is carefully designed. Specifically, we randomly sample true positive pairs from source and target domain in VisDA-2017 dataset (Peng et al., 2017) as the training data. Then we manually replace the true positive pairs with random false positive pairs to increase the noise, and watch the changes of the performance as shown in Fig. 1b. The x-axis indicates the rate of false positive pairs in the training data, and the y-axis shows the performance of different methods on the UDA task. The red curve represents the results by aligning pairs with the cross-attention module, while the green curve is that without cross-attention, *i.e.* to directly train the target data with the label of corresponding source data in the pair. It can be seen that the red curve achieves a much better performance than the green one, which implies the robustness of the cross-attention module to the noise. We also provide another baseline shown as the blue curve in Fig. 1b, which is to remove the false positive pairs from the training data and train the cross-attention with only true positive pairs. Without the noisy data, this baseline can be considered as the upper bound to our methods. We can see the red curve is very close to the blue curve, and both of them are much better than the green one. It further implies that the cross-attention module is robust to the noisy input pair.

---

[1]The patches in true positive pairs, either similar and dissimilar, would bring no noise to the final performance, which is out of the discussion of our paper.

### 3.2 TWO-WAY CENTER-AWARE PSEUDO LABELING

#### 3.2.1 TWO-WAY LABELING

To build the training pairs for the cross-attention module, an intuitive method is that for each image in the source domain, we manage to find the most similar image from the target domain. The set $\mathbb{P}_S$ of selected pairs is:

$$\mathbb{P}_S = \{(s,t)|t = \min_k d(\boldsymbol{f}_s, \boldsymbol{f}_k), \forall k \in T, \forall s \in S\} \quad (3)$$

where $S, T$ are the source and target data respectively. $d(\boldsymbol{f}_i, \boldsymbol{f}_j)$ means the distance between features of image $i$ and $j$. The advantage of this strategy is to make full use of source data, while its weakness is obvious that only a part of target data is involved. To eliminate this training bias from target data, we introduce more pairs $\mathbb{P}_T$ from the opposite way, consisting of all the target data and their corresponding most similar images in the source domain.

$$\mathbb{P}_T = \{(s,t)|s = \min_k d(\boldsymbol{f}_t, \boldsymbol{f}_k), \forall t \in T, \forall k \in S\} \quad (4)$$

As a result the final set $\mathbb{P}$ is the union of two sets, *i.e.* $\mathbb{P} = \{\mathbb{P}_S \cup \mathbb{P}_T\}$, making the training pairs include all the source and target data.

#### 3.2.2 CENTER-AWARE FILTERING

The pairs in $\mathbb{P}$ are built based on the feature similarities of images from both domains, thus the accuracy of the pseudo labels of pairs is highly dependent on the feature similarities. Inspired by Liang et al. (2020), we find that the pre-trained model of the source data is also useful to further improve the accuracy. Firstly, we send all the target data through the pre-trained model and obtain their probability distributions $\delta$ on the source categories from the classifier. Similar to Liang et al. (2020), these distributions can be used to compute initial centers of each category in the target domain by weighted k-means clustering:

$$\boldsymbol{c}_k = \frac{\sum_{t \in T} \delta_t^k \boldsymbol{f}_t}{\sum_{t \in T} \delta_t^k} \quad (5)$$

where $\delta_t^k$ indicates the probability of image $t$ on category $k$. Pseudo labels of the target data can be produced via the nearest neighbor classifier:

$$y_t = \arg\min_k d(\boldsymbol{c}_k, \boldsymbol{f}_t) \quad (6)$$

where $t \in T$ and $d(i,j)$ is the distance of features $i$ and $j$. Based on the pseudo labels, we can calculate new centers:

$$\boldsymbol{c}_k' = \frac{\sum_{t \in T} \mathbb{1}(y_t = k)\boldsymbol{f}_t}{\sum_{t \in T} \mathbb{1}(y_t = k)} \quad (7)$$

In Liang et al. (2020), Eq. 6 and 7 could be updated for multiple rounds, and we only adopt one round in our paper. The final pseudo labels are then used to refine the selected pairs. Specifically, for every pair, if the pseudo label of the target image is consistent with the label of the source image, this pair would be kept for our training, otherwise it will be discarded as a noise.

### 3.3 CDTRANS: CROSS-DOMAIN TRANSFORMER

The framework of the proposed Cross-domain Transformer (CDTrans) is shown in Fig. 2, which consists of three weight-sharing transformers. There are three data flows and constraints for the weight-sharing branches.

The inputs of the framework are the selected pairs from our labeling method mentioned above. The three branches are named as *source branch*, *target branch*, *source-target branch*. As shown in Fig. 2, the source and target images in the input pair are sent to source branch and target branch respectively. In these two branches, the self-attention module is involved to learn the domain-specific

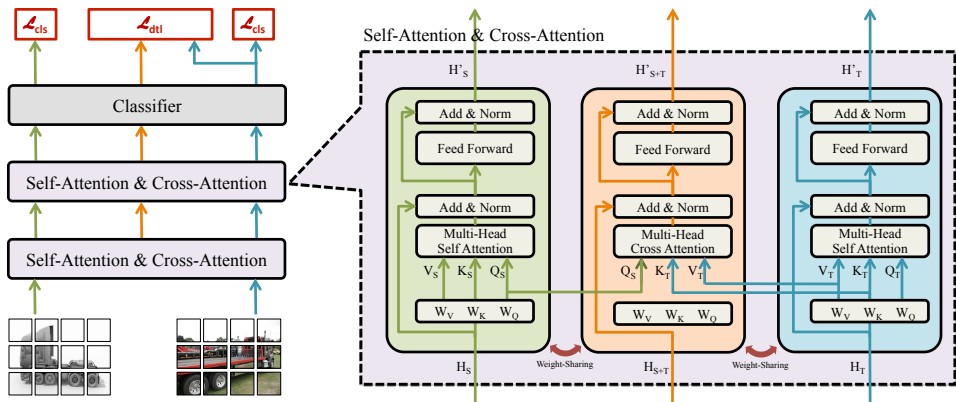

Figure 2: The proposed CDTrans framework. It consists of three weight-sharing transformers fed by inputs from the selected pairs using the two-way center-aware labeling method. Cross-entropy is adopted to *source branch* ($H_S$) and *target branch* ($H_T$), while the distillation loss is applied between *source-target branch* ($H_{S+T}$) and $H_T$.

representations. And the softmax cross-entropy loss is used to train the classification. It is worth noting that all three branches share the same classifier due to the same label of two images.

The cross-attention module is imported in the source-target branch. The inputs of the source-target branch are from the other two branches. In the $N$-th layer, the query of the cross-attention module comes from the query in the $N$-th layer of the source branch, while the keys and values are from those of the target branch. Then the cross-attention module outputs aligned features which are added with the output of the $(N-1)$-th layer.[2]

The features of the source-target branch not only align distributions of two domains, but are robust to the noise in the input pairs thanks to the cross-attention module. Thus we use the output of the source-target branch to guide the training of the target branch. Specifically, the source-target branch and target branch are denoted as teacher and student respectively. We consider the probability distribution of the classifier in source-target branch as a soft label that can be used to further supervise the target branch through a distillation loss (Hinton et al., 2015):

$$L_{dtl} = \sum_k q_k \log p_k \tag{8}$$

where $q_k$ and $p_k$ are the probabilities of category $k$ from the source-target branch and the target branch respectively.

During inference, only the target branch is used. The input is an image from testing data, and only the target data flow is triggered, *i.e.* the blue lines in Fig. 2. Its output of the classifier is utilized as the final predicted labels.

## 4 EXPERIMENTS

### 4.1 DATASETS AND IMPLEMENTATION

The proposed method is verified on four popular UDA benchmarks, including VisDA-2017 (Peng et al., 2017), Office-Home (Venkateswara et al., 2017), Office-31 (Saenko et al., 2010) and DomainNet (Peng et al., 2019). The input image size in our experiments is 224×224. Both the DeiT-small and DeiT-base (Touvron et al., 2021) are adopted as our backbone for fair comparison. We use the Stochastic Gradient Descent algorithm with the momentum of 0.9 and weight decay ratio 1e-4 to optimize the training process. The learning rate is set to 3e-3 for Office-Home, Office-31 and DomainNet, 5e-5 for VisDA-2017 since it can easily converge. The batch size is set to 64.

---

[2]The addition operation is not included in the 1st layer.

| Method | plane | bcycl | bus | car | horse | knife | mcycl | person | plant | sktbrd | train | truck | Avg. |
|---|---|---|---|---|---|---|---|---|---|---|---|---|---|
| ResNet-50 | 55.1 | 53.3 | 61.9 | 59.1 | 80.6 | 17.9 | 79.7 | 31.2 | 81.0 | 26.5 | 73.5 | 8.5 | 52.4 |
| DANN | 81.9 | 77.7 | 82.8 | 44.3 | 81.2 | 29.5 | 65.1 | 28.6 | 51.9 | 54.6 | 82.8 | 7.8 | 57.4 |
| MinEnt | 80.3 | 75.5 | 75.8 | 48.3 | 77.9 | 27.3 | 69.7 | 40.2 | 46.5 | 46.6 | 79.3 | 16.0 | 57.0 |
| MCD | 87.0 | 60.9 | 83.7 | 64.0 | 88.9 | 79.6 | 84.7 | 76.9 | 88.6 | 40.3 | 83.0 | 25.8 | 71.9 |
| SWD | 90.8 | 82.5 | 81.7 | 70.5 | 91.7 | 69.5 | 86.3 | 77.5 | 87.4 | 63.6 | 85.6 | 29.2 | 76.4 |
| CDAN+E | 85.2 | 66.9 | 83.0 | 50.8 | 84.2 | 74.9 | 88.1 | 74.5 | 83.4 | 76.0 | 81.9 | 38.0 | 73.9 |
| BNM | 89.6 | 61.5 | 76.9 | 55.0 | 89.3 | 69.1 | 81.3 | 65.5 | 90.0 | 47.3 | 89.1 | 30.1 | 70.4 |
| MSTN+DSBN | 94.7 | 86.7 | 76.0 | 72.0 | 95.2 | 75.1 | 87.9 | 81.3 | 91.1 | 68.9 | 88.3 | 45.5 | 80.2 |
| CGDM | 93.7 | 82.7 | 73.2 | 68.4 | 92.9 | 94.5 | 88.7 | 82.1 | 93.4 | 82.5 | 86.8 | 49.2 | 82.3 |
| CGDM* | 92.8 | 85.1 | 76.3 | 64.5 | 91.0 | 93.2 | 81.3 | 79.3 | 92.4 | 83.0 | 85.6 | 44.8 | 80.8 |
| SHOT | 94.3 | 88.5 | 80.1 | 57.3 | 93.1 | 93.1 | 80.7 | 80.3 | 91.5 | 89.1 | 86.3 | 58.2 | 82.9 |
| SHOT* | 95.5 | 87.5 | 80.1 | 54.5 | 93.6 | 94.2 | 80.2 | 80.9 | 90.0 | 89.9 | 87.1 | 58.4 | 82.7 |
| TVT° | 92.9 | 85.6 | 77.5 | 60.5 | 93.6 | 98.2 | 89.4 | 76.4 | 93.6 | **92.0** | 91.7 | 55.7 | 83.9 |
| Baseline-B | 97.7 | 48.1 | 86.6 | 61.6 | 78.1 | 63.4 | **94.7** | 10.3 | 87.7 | 47.7 | **94.4** | 35.5 | 67.1 |
| CGDM-B* | 96.3 | 87.1 | **86.8** | **83.5** | 92.2 | 98.3 | 91.6 | 78.5 | 96.3 | 48.4 | 89.4 | 39.0 | 82.3 |
| SHOT-B* | **97.9** | 90.3 | 86.0 | 73.4 | **96.9** | **98.8** | 94.3 | 54.8 | 95.4 | 87.1 | 93.4 | 62.7 | 85.9 |
| Ours-B | 97.1 | **90.5** | 82.4 | 77.5 | 96.6 | 96.1 | 93.6 | **88.6** | **97.9** | 86.9 | 90.3 | **62.8** | **88.4** |

Table 1: Comparison with SoTA methods on VisDA-2017. "S/B" implies the DeiT-small/DeiT-base backbone respectively. ∗ indicates the results are reproduced by ourselves. ◦ implies its pretrained model is trained on ImageNet21K instead of ImageNet1K. The best performance is marked as **bold**.

| Method | Ar→Cl | Ar→Pr | Ar→Re | Cl→Ar | Cl→Pr | Cl→Re | Pr→Ar | Pr→Cl | Pr→Re | Re→Ar | Re→Cl | Re→Pr | Avg. |
|---|---|---|---|---|---|---|---|---|---|---|---|---|---|
| ResNet-50 | 44.9 | 66.3 | 74.3 | 51.8 | 61.9 | 63.6 | 52.4 | 39.1 | 71.2 | 63.8 | 45.9 | 77.2 | 59.4 |
| MinEnt | 51.0 | 71.9 | 77.1 | 61.2 | 69.1 | 70.1 | 59.3 | 48.7 | 77.0 | 70.4 | 53.0 | 81.0 | 65.8 |
| CDAN+E | 54.6 | 74.1 | 78.1 | 63.0 | 72.2 | 74.1 | 61.6 | 52.3 | 79.1 | 72.3 | 57.3 | 82.8 | 68.5 |
| DCAN | 54.5 | 75.7 | 81.2 | 67.4 | 74.0 | 76.3 | 67.4 | 52.7 | 80.6 | 74.1 | 59.1 | 83.5 | 70.5 |
| BNM | 56.7 | 77.5 | 81.0 | 67.3 | 76.3 | 77.1 | 65.3 | 55.1 | 82.0 | 73.6 | 57.0 | 84.3 | 71.1 |
| ATDOC-NA | 58.3 | 78.8 | 82.3 | 69.4 | 78.2 | 78.2 | 67.1 | 56.0 | 82.7 | 72.0 | 58.2 | 85.5 | 72.2 |
| SHOT | 57.1 | 78.1 | 81.5 | 68.0 | 78.2 | 78.1 | 67.4 | 54.9 | 82.2 | 73.3 | 58.8 | 84.3 | 71.8 |
| SHOT* | 56.2 | 77.6 | 81.6 | 67.5 | 78.2 | 78.8 | 67.8 | 54.0 | 82.0 | 72.5 | 58.8 | 84.5 | 71.6 |
| TVT° | **74.9** | **86.8** | **89.5** | **82.8** | **88.0** | **88.3** | **79.8** | **71.9** | **90.1** | **85.5** | **74.6** | 90.6 | **83.6** |
| Baseline-S | 55.6 | 73.0 | 79.4 | 70.6 | 72.9 | 76.3 | 67.5 | 51.0 | 81.0 | 74.5 | 53.2 | 82.7 | 69.8 |
| Ours-S | 60.6 | 79.5 | 82.4 | 75.6 | 81.0 | 82.3 | 72.5 | 56.7 | 84.4 | 77.0 | 59.1 | 85.5 | 74.7 |
| Baseline-B | 61.8 | 79.5 | 84.3 | 75.4 | 78.8 | 81.2 | 72.8 | 55.7 | 84.4 | 78.3 | 59.3 | 86.0 | 74.8 |
| CGDM-B* | 67.1 | 83.9 | 85.4 | 77.2 | 83.3 | 83.7 | 74.6 | 64.7 | 85.6 | 79.3 | 69.5 | 87.7 | 78.5 |
| SHOT-B* | 67.1 | 83.5 | 85.5 | 76.6 | 83.4 | 83.7 | 76.3 | 65.3 | 85.3 | 80.4 | 66.7 | 83.4 | 78.1 |
| Ours-B | 68.8 | 85.0 | 86.9 | 81.5 | 87.1 | 87.3 | 79.6 | 63.3 | 88.2 | 82.0 | 66.0 | **90.6** | 80.5 |

Table 2: Comparison with SoTA methods on Office-Home. The best performance is marked as **bold**.

## 4.2 COMPARISON TO SoTA

We compare our method with state-of-the-art methods on UDA tasks, including MinEnt (Grandvalet et al., 2005), DAN (Long et al., 2015), DANN (Ganin & Lempitsky, 2015), CDAN+E (Long et al., 2018), CDAN+BSP (Chen et al., 2019), CDAN+TN (Wang et al., 2019), rRGrad+CAT (Deng et al., 2019), MCD (Saito et al., 2018), SWD (Lee et al., 2019), MSTN+DSBN (Chang et al., 2019), SAFN+ENT (Xu et al., 2019), BNM (Cui et al., 2020), DCAN (Li et al., 2020), SHOT (Liang et al., 2020), ATDOC-NA (Liang et al., 2021), CGDM (Du et al., 2021) and TVT (Yang et al.). The results are shown in Table 1, 2, 3 and 4.

For Office-Home, Office-31 and DomainNet, as most of the methods use ResNet-50 as their backbones, we provide results with DeiT-small as our backbone for a fair comparison, which has a comparable model size as ResNet-50, but we also show the results using DeiT-base. And for VisDA-2017, we adopt the DeiT-base backbone for fair comparisons, where other methods utilize ResNet-101 for their results.

The "Baseline-S/B" indicates directly training a DeiT-small/DeiT-base on the source domain and testing on the target domain. The baseline shows a competitive result even compared to other SoTA methods on most datasets. It demonstrates that Transformers has better generalization ability over ConvNets. We also provide some insights about why transformers can generalize well from source domain to target domain in the supplementary materials. To further eliminate the unfairness of using different backbones, we reproduce the results of SHOT and CGDM (marked as "*"), and replace their backbones with DeiT-base as the same as ours, denoted as "-B*".

| MCD | clp | inf | pnt | qdr | rel | skt | Avg. |
|---|---|---|---|---|---|---|---|
| clp | - | 15.4 | 25.5 | 3.3 | 44.6 | 31.2 | 24.0 |
| inf | 24.1 | - | 24.0 | 1.6 | 35.2 | 19.7 | 20.9 |
| pnt | 31.1 | 14.8 | - | 1.7 | 48.1 | 22.8 | 23.7 |
| qdr | 8.5 | 2.1 | 4.6 | - | 7.9 | 7.1 | 6.0 |
| rel | 39.4 | 17.8 | 41.2 | 1.5 | - | 25.2 | 25.0 |
| skt | 37.3 | 12.6 | 27.2 | 4.1 | 34.5 | - | 23.1 |
| Avg. | 28.1 | 12.5 | 24.5 | 2.4 | 34.1 | 21.2 | 20.5 |

| CDAN | clp | inf | pnt | qdr | rel | skt | Avg. |
|---|---|---|---|---|---|---|---|
| clp | - | 13.5 | 28.3 | 9.3 | 43.8 | 30.2 | 25.0 |
| inf | 18.9 | - | 21.4 | 1.9 | 36.3 | 21.3 | 20.0 |
| pnt | 29.6 | 14.4 | - | 4.1 | 45.2 | 27.4 | 24.2 |
| qdr | 11.8 | 1.2 | 4.0 | - | 9.4 | 9.5 | 7.2 |
| rel | 36.4 | 18.3 | 40.9 | 3.4 | - | 24.6 | 24.7 |
| skt | 38.2 | 14.7 | 33.9 | 7.0 | 36.6 | - | 26.1 |
| Avg. | 27.0 | 12.4 | 25.7 | 5.1 | 34.3 | 22.6 | 21.2 |

| BNM | clp | inf | pnt | qdr | rel | skt | Avg. |
|---|---|---|---|---|---|---|---|
| clp | - | 12.1 | 33.1 | 6.2 | 50.8 | 40.2 | 28.5 |
| inf | 26.6 | - | 28.5 | 2.4 | 38.5 | 18.1 | 22.8 |
| pnt | 39.9 | 12.2 | - | 3.4 | 54.5 | 36.2 | 29.2 |
| qdr | 17.8 | 1.0 | 3.6 | - | 9.2 | 8.3 | 8.0 |
| rel | 48.6 | 13.2 | 49.7 | 3.6 | - | 33.9 | 29.8 |
| skt | 54.9 | 12.8 | 42.3 | 5.4 | 51.3 | - | 33.3 |
| Avg. | 37.6 | 10.3 | 31.4 | 4.2 | 40.9 | 27.3 | 25.3 |

| SWD | clp | inf | pnt | qdr | rel | skt | Avg. |
|---|---|---|---|---|---|---|---|
| clp | - | 14.7 | 31.9 | 10.1 | 45.3 | 36.5 | 27.7 |
| inf | 22.9 | - | 24.2 | 2.5 | 33.2 | 21.3 | 20.0 |
| pnt | 33.6 | 15.3 | - | 4.4 | 46.1 | 30.7 | 26.0 |
| qdr | 15.5 | 2.2 | 6.4 | - | 11.1 | 10.2 | 9.1 |
| rel | 41.2 | 18.1 | 44.2 | 4.6 | - | 31.6 | 27.9 |
| skt | 44.2 | 15.2 | 37.3 | 10.3 | 44.7 | - | 30.3 |
| Avg. | 31.5 | 13.1 | 28.8 | 6.4 | 36.1 | 26.1 | 23.6 |

| CGDM | clp | inf | pnt | qdr | rel | skt | Avg. |
|---|---|---|---|---|---|---|---|
| clp | - | 16.9 | 35.3 | 10.8 | 53.5 | 36.9 | 30.7 |
| inf | 27.8 | - | 28.2 | 4.4 | 48.2 | 22.5 | 26.2 |
| pnt | 37.7 | 14.5 | - | 4.6 | 59.4 | 33.5 | 30.0 |
| qdr | 14.9 | 1.5 | 6.2 | - | 10.9 | 10.2 | 8.7 |
| rel | 49.4 | 20.8 | 47.2 | 4.8 | - | 38.2 | 32.0 |
| skt | 50.1 | 16.5 | 43.7 | 11.1 | 55.6 | - | 35.4 |
| Avg. | 36.0 | 14.0 | 32.1 | 7.1 | 45.5 | 28.3 | 27.2 |

| Base-S | clp | inf | pnt | qdr | rel | skt | Avg. |
|---|---|---|---|---|---|---|---|
| clp | - | 21.2 | 44.2 | 15.3 | 59.9 | 46.0 | 37.3 |
| inf | 36.8 | - | 39.4 | 5.4 | 52.1 | 32.6 | 33.3 |
| pnt | 47.1 | 21.7 | - | 5.7 | 60.2 | 39.9 | 34.9 |
| qdr | 25.0 | 3.3 | 10.4 | - | 18.8 | 14.0 | 14.3 |
| rel | 54.8 | 23.9 | 52.6 | 7.4 | - | 40.1 | 35.8 |
| skt | 55.6 | 18.6 | 42.7 | 14.9 | 55.7 | - | 37.5 |
| Avg. | 43.9 | 17.7 | 37.9 | 9.7 | 49.3 | 34.5 | 32.2 |

| Ours-S | clp | inf | pnt | qdr | rel | skt | Avg. |
|---|---|---|---|---|---|---|---|
| clp | - | 25.3 | 52.5 | 23.2 | 68.3 | 53.2 | 44.5 |
| inf | 47.6 | - | 48.3 | 9.9 | 62.8 | 41.1 | 41.9 |
| pnt | 55.4 | 24.5 | - | 11.7 | 67.4 | 48.0 | 41.4 |
| qdr | 36.6 | 5.3 | 19.3 | - | 33.8 | 22.7 | 23.5 |
| rel | 61.5 | 28.1 | 56.8 | 12.8 | - | 47.2 | 41.3 |
| skt | 64.3 | 26.1 | 53.2 | 23.9 | 66.2 | - | 46.7 |
| Avg. | 53.1 | 21.9 | 46.0 | 16.3 | 59.7 | 42.4 | 39.9 |

| Base-B | clp | inf | pnt | qdr | rel | skt | Avg. |
|---|---|---|---|---|---|---|---|
| clp | - | 24.2 | 48.9 | 15.5 | 63.9 | 50.7 | 40.6 |
| inf | 43.5 | - | 44.9 | 6.5 | 58.8 | 37.6 | 38.3 |
| pnt | 52.8 | 23.3 | - | 6.6 | 64.6 | 44.5 | 38.4 |
| qdr | 31.8 | 6.1 | 15.6 | - | 23.4 | 18.9 | 19.2 |
| rel | 58.9 | 26.3 | 56.7 | 9.1 | - | 45.0 | 39.2 |
| skt | 60.0 | 21.1 | 48.4 | 16.6 | 61.7 | - | 41.6 |
| Avg. | 49.4 | 20.2 | 42.9 | 10.9 | 54.5 | 39.3 | 36.2 |

| Ours-B | clp | inf | pnt | qdr | rel | skt | Avg. |
|---|---|---|---|---|---|---|---|
| clp | - | 29.4 | 57.2 | 26.0 | 72.6 | 58.1 | 48.7 |
| inf | 57.0 | - | 54.4 | 12.8 | 69.5 | 48.4 | 48.4 |
| pnt | 62.9 | 27.4 | - | 15.8 | 72.1 | 53.9 | 46.4 |
| qdr | 44.6 | 8.9 | 29.0 | - | 42.6 | 28.5 | 30.7 |
| rel | 66.2 | 31.0 | 61.5 | 16.2 | - | 52.9 | 45.6 |
| skt | 69.0 | 29.6 | 59.0 | 27.2 | 72.5 | - | 51.5 |
| Avg. | 59.9 | 25.3 | 52.2 | 19.6 | 65.9 | 48.4 | **45.2** |

Table 4: Comparison with SoTA methods on DomainNet. "Base" is the Baseline.

From Table 1, 2, 3 and 4, it can be seen that our method outperforms the baseline with a large margin on all four datasets, *e.g.* nearly 21% on VisDA. With our improvements, the new Transformer with cross-attention module shows a much better generalization power, and achieves the best performance on VisDA-2017 compared to other SoTAs methods. It further implies the effectiveness of our method on the UDA task.

Taking a closer look at the results, for the hard categories, such as "person" in VisDA-2017 dataset, the baseline is very low, which indicates the initial model of our method has a poor classification ability on this category, leading to the pseudo labels with more noise. Even with such a poor baseline and poor quality of pseudo labels, our method can still achieve a much higher performance boost (from 10.3% to 88.6%). It suggests that our method has a great robustness to the labeling noise and can overcome the noise problem to some extent.

| Method | A→D | A→W | D→A | D→W | W→A | W→D | Avg |
|---|---|---|---|---|---|---|---|
| ResNet-50 | 68.9 | 68.4 | 62.5 | 96.7 | 60.7 | 99.3 | 76.1 |
| DANN | 79.7 | 82.0 | 68.2 | 96.9 | 67.4 | 99.1 | 82.2 |
| CDAN+E | 92.9 | 94.1 | 71.0 | 98.6 | 69.3 | 100. | 87.7 |
| rRGrad+CAT | 90.8 | 94.4 | 72.2 | 98.0 | 70.2 | 100. | 87.6 |
| SAFN+ENT | 90.7 | 90.1 | 73.0 | 98.6 | 70.2 | 99.8 | 87.1 |
| CDAN+BSP | 93.0 | 93.3 | 73.6 | 98.2 | 72.6 | 100. | 88.5 |
| CDAN+TN | 94.0 | 95.7 | 73.4 | 98.7 | 74.2 | 100. | 89.3 |
| SHOT | 94.0 | 90.1 | 74.7 | 98.4 | 74.3 | 99.9 | 88.6 |
| SHOT* | 93.8 | 91.8 | 74.8 | 98.2 | 74.1 | 99.8 | 88.8 |
| TVT° | 96.4 | 96.4 | **84.9** | **99.4** | **86.1** | 100. | **93.8** |
| Baseline-S | 87.6 | 86.9 | 74.9 | 97.7 | 73.5 | 99.6 | 86.7 |
| Ours-S | 94.6 | 93.5 | 78.4 | 98.2 | 78.0 | 99.6 | 90.4 |
| Baseline-B | 90.8 | 90.4 | 76.8 | 98.2 | 76.4 | 100. | 88.8 |
| CGDM-B* | 94.6 | 95.3 | 78.8 | 97.6 | 81.2 | 99.8 | 91.2 |
| SHOT-B* | 95.3 | 94.3 | 79.4 | 99.0 | 80.2 | 100. | 91.4 |
| Ours-B | **97.0** | **96.7** | 81.1 | 99.0 | 81.9 | 100. | 92.6 |

Table 3: Comparison with SoTA methods on Office-31. The best performance is marked as **bold**.

We can see that TVT achieves a better result on Office-Home and Office-31. Because TVT utilizes ViT (Dosovitskiy et al., 2020) as backbone which is pretrained on ImageNet21K. While the pretrained model of our CDTrans and other UDA methods are trained on ImageNet1K.

## 4.3 ABLATION STUDY

### 4.3.1 DIFFERENT PSEUDO LABELING

We have conducted experiments on different pseudo labeling methods to verify their influence on the final performance. The results on VisDa-2017 are listed in Table. 5. RPLL (Zheng & Yang, 2021) and MRKLD+LRENT (Zou et al., 2019) are two commonly used pseudo-label generation methods, we reproduce their pseudo-label generations on our baseline to compare with our proposed pseudo labeling method. $Rec_s$, $Rec_t$ means the recall of the selected training pairs in the source and target data, while $Prec$ represents the accuracy of the pairs. "One-way-source" and "One-way-target" denote only using the pair set $\mathbb{P}_S$ in Eq. 3 or $\mathbb{P}_T$ in Eq. 4 for training. "Two-way" indicates results

| Pseudo labels | $Rec_s$ | $Rec_t$ | Prec | plane | bcycl | bus | car | horse | knife | mcycl | person | plant | sktbrd | train | truck | Avg. |
|---|---|---|---|---|---|---|---|---|---|---|---|---|---|---|---|---|
| One-way-source | 100. | 6.6 | 90.6 | 96.1 | 52.7 | 85.5 | 69.6 | 95.0 | 90.2 | 95.1 | 66.6 | 88.8 | 54.6 | 95.4 | 29.5 | 76.6 |
| One-way-target | 8.0 | 100. | 76.3 | 98.2 | 32.0 | 87.7 | 84.1 | 95.5 | 89.9 | 98.3 | 66.8 | 95.7 | 57.5 | 95.6 | 22.0 | 76.9 |
| Two-way | 100. | 100. | 81.8 | 97.5 | 49.6 | 88.7 | 73.9 | 94.6 | 85.8 | 96.6 | 58.6 | 93.3 | 63.6 | 94.8 | 27.9 | 77.1 |
| Tw + Ca | 97.8 | 94.8 | 91.3 | 98.1 | 86.9 | 87.9 | 80.9 | 97.9 | 97.3 | 96.8 | 85.3 | 97.6 | 83.2 | 94.0 | 54.4 | 88.4 |
| RPLL | - | - | - | 98.4 | 63.4 | 85.8 | 68.8 | 97.0 | 95.4 | 97.77 | 59.3 | 96.2 | 57.2 | 96.2 | 48.1 | 80.3 |
| MRKLD+LRENT | - | - | - | 97.8 | 77.3 | 81.4 | 64.3 | 94.6 | 93.9 | 93.3 | 77.5 | 93.1 | 74.9 | 92.6 | 59.0 | 83.3 |
| Groundtruth | 100. | 100. | 100. | 97.9 | 89.1 | 92.3 | 91.9 | 98.4 | 97.2 | 97.5 | 86.8 | 98.6 | 90.7 | 96.3 | 60.0 | 91.5 |

Table 5: Comparison among different pseudo labeling methods on VisDa-2017. $Rec_s$, $Rec_t$ express the recall of pseudo labels in source and target data, while $Prec$ represents the accuracy of the pairs. "One-way-source/target" denotes only using the source/target pair set for training. "Tw+Ca" implies the proposed two-way center-aware labeling method.

| $L_s$ | $L_{s+t}$ | $L_t$ | plane | bcycl | bus | car | horse | knife | mcycl | person | plant | sktbrd | train | truck | Avg. |
|---|---|---|---|---|---|---|---|---|---|---|---|---|---|---|---|
| cls | - | - | 97.7 | 48.1 | 86.6 | 61.6 | 78.1 | 63.4 | 94.7 | 10.3 | 87.7 | 47.7 | 94.4 | 35.5 | 67.1 |
| - | - | cls | 98.3 | 85.0 | 88.0 | 76.3 | 98.1 | 96.1 | 96.9 | 61.1 | 97.2 | 85.5 | 94.6 | 54.9 | 86.0 |
| cls | - | cls | 98.3 | 87.4 | 89.1 | 77.3 | 98.0 | 97.4 | 95.4 | 69.5 | 97.1 | 86.3 | 95.3 | 49.5 | 86.7 |
| cls | cls | cls | 98.2 | 88.4 | 88.0 | 76.8 | 98.2 | 97.2 | 95.6 | 80.1 | 97.1 | 84.7 | 94.5 | 54.1 | 87.7 |
| cls | dtl | cls | 98.0 | 86.9 | 87.9 | 80.9 | 97.9 | 97.3 | 96.8 | 85.3 | 97.6 | 83.2 | 94.0 | 54.4 | 88.4 |

Table 6: Comparison among different losses on VisDa-2017. $L_s$, $L_t$ and $L_{s+t}$ represent the loss used in source, target and source+target branches respectively. $cls$ and $dtl$ imply the classification loss and the distillation loss.

of using the union of $\mathbb{P}_s$ and $\mathbb{P}_t$ without the center-aware strategy. "Tw+Ca" implies our two-way center-aware labeling method, and "Groundtruth" means all training pairs are from groundtruthes.

By looking at $Rec_s$ and $Rec_t$ in Table. 5, it can be found that the one-way methods have an apparent bias on either source or target data, and its results are lower than the two-way method. By comparing "Two-way" and "Tw+Ca", we can conclude that although the center-aware method filters the training pair and slightly reduces the recall, it largely improves the precision and leads to a better final performance. We also find that our two-way center-aware labeling method achieves a very high result, not only better than other pseudo-label generation methods, but also very close to the upper bound trained with groundtruth pairs.

### 4.3.2 Different Losses

As there are three losses in our method, we conduct another experiment to verify the effectiveness of each loss on VisDa-2017, as shown in Table. 6. "cls" in $L_{s+t}$ denotes that we replace the distillation loss with a classification loss for the source-target branch. We can see that the 3rd row with both $L_s$ and $L_t$ having classification loss achieves a better result than the first row where only $L_s$ has the cls loss, which means the target branch with the pseudo labels is helpful to improve the UDA result. With the addition of "cls" in $L_{s+t}$, the performance is further improved, which demonstrates the advantages of using the cross-attention module for feature alignment. Using "dtl" instead of "cls" on the source-target branch can further improve the results, showing the effectiveness of our distillation loss.

## 5 Conclusion

In this paper, we tackle the problem of unsupervised domain adaptation by introducing the cross-attention module into Transformer in a novel way. We propose a new network structure CDTrans which is a pure transformer-based structure with three branches, and we also propose to generate high-quality pseudo labels using a two-way center-aware labeling method. Training CDTrans using the generated high-quality pseudo labels yields a robust solution and also achieves state-of-the-art results on four popular UDA datasets, outperforming previous methods by a large margin. We believe that transformer-based approaches will have great potential in the UDA community, and our work, as one of the first attempts along this direction, has pushed forward the frontiers and shed lights for future research.

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
