# OpenReview forum: "CDTrans: Cross-domain Transformer for Unsupervised Domain Adaptation"
_ICLR.cc/2022/Conference — ICLR 2022 Poster_

### Official Review · Reviewer_1DYQ · 2021-10-28

**Correctness:** 3
**Technical Novelty And Significance:** 2
**Empirical Novelty And Significance:** 3
**Recommendation:** 6
**Confidence:** 3

**Main Review:**

This paper has the following strengths.
- The problem addressed is interesting for the community. The study of pseudo labeling is under-explored compared with domain invariant feature learning.
- The figures describing the different modules are clear and stimulating.
- The method achieves outstanding performance on multiple datasets.

My main concerns are as follows:
- Comparison with other methods is unfair. From the experiments, we can clearly see the superiority of the adopted Transformer-based backbone against ResNet. Baseline-B even surpasses most previous works under the source-only setting. However, most previous works compared (e.g, SHOT, CGDM) use ResNet as the backbone. To better show the effectiveness of the proposed method, it is better to adopt the domain adaptation techniques of other works with the same backbone.
- The technical contribution of this work is limited. There are many works that have used cross transformers in computer vision and NLP tasks. However, not specifically in the UDA research landscape.

**Summary Of The Paper:**

In this work, the authors propose a method for domain adaptation by introducing a new way of generating pseudo labels and a cross-transformer with classification and distillation losses. The authors used a cross-transformer where the queries come from the source domain and the values and keys come from the target domain, and try to minimize the output distribution difference between the cross branch and the target branch by a distillation loss. Since the transformers compare source and target in a patch-based manner, the authors find it is more robust to false-positive pairs. The authors do not focus on learning domain-invariant feature representation and mainly solve the domain adaptation problem by pseudo labeling. In four public datasets, the proposed method showed outstanding performance.

**Summary Of The Review:**

Despite the concern on novelty, overall this paper lacks justification of their method to see whether the improvement is brought by the backbone change. The authors should compare other methods when replaced with the same backbone as the proposed method.

---

> ### Author Response · Authors · 2021-11-19
> **Response to Reviewer 1DYQ**
>
> **Q1. [Compared SoTAs with the same backbone]** Per your suggestion, we have reproduced the results of SHOT and CGDM by replacing their backbones with DeiT-base backbone, the same with ours. The new results are listed in Table 1,2,3. It can be seen that their results with DeiT-base are better than the transformer baseline, which implies the effectiveness of their methods. Our method still achieves better performance than them even with the same backbone. The DomainNet includes 30 sets of experiments which are too time consuming to be finished before Nov.22. We will add it into the paper as soon as the results come out, although we believe the comparisons on VisDA-2017, Office-home and Office-31 are enough to imply the effectiveness of our method. It is worth noting that the open source implementation of CGDM cannot be executed out-of-the-box because of some missing functions. We tried our best to fix those problems to reproduce their results; however, the reproduced results are slight worse than the numbers reported in the original paper in VisDA-2017. A possible explanation is that the parameters in CGDM need to be carefully tuned.
>
> **Q2. [Technical contribution]** Yes, cross transformers have been used in computer vision and NLP tasks, especially in multimodal tasks. However, most of these cross-transformer-based methods try to perform feature alignment/fusion supervised by the groundtruth labels. Different from them, in the UDA task, there are no labels in the target domain, and how to effectively adopt cross transformers is a nontrivial problem. We demonstrate, both practically and theoretically, that cross attention is robust to noise  (the details can be found in Section 3.1.2, and the theoretical analysis is added in the supplemental material). The proposed two-way center-away pseudo labeling method benefit from this robustness. And our method is the first work and a novel work to apply cross-attention for UDA.

---

> > ### Public Comment · ~Glen_Fowler1 · 2022-10-03
> > **Thanks**
> >
> > Thanks for the info.

---

### Official Review · Reviewer_3J2D · 2021-11-01

**Correctness:** 4
**Technical Novelty And Significance:** 4
**Empirical Novelty And Significance:** 4
**Recommendation:** 8
**Confidence:** 4

**Main Review:**

Strengths:
-- an exploration about cross attention layer is conducted and it is discovered that that the cross attention layer is robust to pseudo label noise.

-- a novel three branches architecture is proposed in this submission, and the source-target branch is utilized as a teacher to guide the other two branches during training. As far as I know, no similar work was proposed before on UDA classification tasks.

-- the experimental results seem promising, which demonstrate the efficacy of this method.

Weakness:
-- I am curious about the memory cost and the model size of the propose method. And inference speed is also expected.

**Summary Of The Paper:**

This submission proposes a transformer framework for unsupervised domain adaptive classification tasks. In this submission, they conduct an exploration about cross attention layer and found that the cross attention layer is robust to pseudo label noise. Inspired by this, they construct a three branches architecture in this submission, which includes a source transformer, target transformer, a source-target transformer. Due to the robustness of the cross attention layer, the source-target transformer acts as a teacher to guide the other two branches.

**Summary Of The Review:**

This submission proposes a transformer framework for unsupervised domain adaptive classification tasks. In this submission, they conduct an exploration about cross attention layer and found that the cross attention layer is robust to pseudo label noise. Inspired by this, they construct a three branches architecture in this submission, which includes a source transformer, target transformer, a source-target transformer. Due to the robustness of the cross attention layer, the source-target transformer acts as a teacher to guide the other two branches.

As far as I know, no similar work was proposed before on UDA classification tasks. This work is of novelty and the experimental results are strong enough to demonstrate the efficacy of the proposed method.

---

> ### Author Response · Authors · 2021-11-19
> **Response to Reviewer 3J2D**
>
> **Q1. [Model size and Inference time]** Thanks for your appreciation of our paper. The model size of our method is 86.5M, the same as the baseline, because the parameters in all three branches are shared. The GPU memory of our method and the baseline used during training are 26.3G and 10.9G respectively. As the inputs of our methods are pairs from source and target domain, the batchsize of our method doubles compared to the baseline. The GPU memory cost during inference is 3.7G, and the inference speed is 85.3 img/sec, both the same as the baseline. The reason is that, the input of both our method and the baseline is only one image at a time during inference. **We will definitely open-source the code anonymously within a month, and help other researchers on reproducing our results.**

---

### Official Review · Reviewer_GSc9 · 2021-11-04

**Correctness:** 3
**Technical Novelty And Significance:** 3
**Empirical Novelty And Significance:** 3
**Recommendation:** 8
**Confidence:** 4

**Main Review:**

+ The paper is well-written and easy to follow.
+ The methods achieve SOTA performances in various UDA benchmarks.
+ It's interesting to see the performances of transformers in UDA.
--------------------
- The source only transformers already achieve good baseline UDA performances which already outperforms some of other previous works. But the authors did not provide insights about why transformers has better generalization ability over ConvNets.
- Although the authors provide some ablation study over different pseudo-label generation methods, it seems it is just compared with the two-way's variants in source/target domain. The authors did not provide comparison with the pseudo-labeling methods as follows:
1. Rectifying Pseudo Label Learning via Uncertainty Estimation for Domain Adaptive Semantic Segmentation, IJCV 2020
2. Confidence Regularized Self-Training, ICCV 2019
3. Two-phase Pseudo Label Densification for Self-training based Domain Adaptation, ECCV 2020
I suggest the authors to compare their methods to other pseudo-label generation method to see if the proposed pseudo-label generation improves over them.
- Lack a relevant reference of transformer based UDA:
TVT: Transferable Vision Transformer for Unsupervised Domain Adaptation, arxiv.
(Although it is an arxiv paper, it's also good to include it as it provides better UDA performances on office-31/Home)

**Summary Of The Paper:**

This paper proposes a weight-sharing triple-branch transformer framework, or CDTrans for unsupervised domain adaptation. A two-way center-aware labeling method is proposed to provide better pseudo-labels. SOTA performances were achieved via the proposed method.

**Summary Of The Review:**

I hope the authors could provide some insights/intuition why transformer can generalize well from source to target. And the comparison with other pseudo-labeling approaches is still important to evaluate the contribution.

---

> ### Author Response · Authors · 2021-11-19
> **Response to Reviewer GSc9**
>
> **Q1. [Transformer's generalization ability]** Good question. It's a critical point and we are still working on it. One of the possible reasons is that Transformer is more resilient than CNN to random perturbations made to individual image patches. This is because through the self-attention/cross-attention mechanism, each patch will be combined with all visual similar patches within the same image to form its representation. This combination and weighted averaging process, despite its simplicity, allows us to reduce the impact of noise, like most averaging processes in statistics. To make our argument more rigorous, we include some theoretical analysis in the supplementary materials that reveals the power of averaging in self-attention in terms of reducing noise.
>
> **Q2. [Comparison with other pseudo-label methods]** Per your suggestion, we have reproduced the results of [1,2] and replace our pseudo-labeling generation method with their pseudo-label generation for a fair comparison. The comparison is added into Table 5, from which we can see that both [1] and [2] can achieve better performance than the one-way-target baseline, but still worse than our two-way pseudo-labeling method. It further suggests the effectiveness of our proposed pseudo-labeling method. Reference [3] involves an adversarial discriminator, without the open source implementation, it is difficult for us to adapt it to our baseline during such short time period.
>
> **Q3. [Reference of TVT]** Thanks for your suggestion. TVT is a concurrent work to ours, and it adopts ViT as backbone which is pretrained on ImageNet21K. However, our CDTrans and other UDA methods use DeiT and ResNet as backbones, which are pretrained on ImageNet1K. This explains why TVT achieves better results than ours on small training datasets, such as office-31 and office Home. As suggested by reviewer, we have included the comparisons with TVT in our revision. It's worth mentioning that when we use ImageNet21K pretrained ViT as our backbone, we can achieve 94.0% on Office-31, which is actually better than TVT under a fair comparison.

---

> > ### Comment · Reviewer_GSc9 · 2021-11-29
> > **Re: Response to Reviewer GSc9**
> >
> > Thanks a lot for your detailed responses.
> >
> > 1. I appreciate the efforts in comparing pseudo-labels for RPLL and MRKLD + LRENT and see the improvement brought by two-way and center aware filtering. As the proposed pseudo-label generation can be regarded as an ensemble of two methods, it's also interesting to see other combinations such as MRKLD + LREND plus center aware filtering, etc. BTW, it looks like your MRKLD + LRENT in Visda-17 outperforms the scores in original paper. Would you clarify the reason?
> >
> > 2. For the explanation about why transformers provide better generalization ability than CNNs, I still have some questions. If you are claiming the averaging could help robustness to random perturbation, do you think a simple self-attention module added to CNNs could help, or even outperforming transformers which are patch based methods? Moreover, CNNs will also output averaged output after each convolution layer, if averaging could really help improve generation, do you think increasing the conv kernels' sizes?
> >
> > Lastly, I still feel the explanation is not so satisfactory. Do you think the robustness comes from the patch based approach? Patches might more resilient to pixel-level perturbation. Moreover, shapes might be important for domain adaptation. Patches are mid-level units between pixels and whole images, which might be better in capturing the shapes and generalize better.
> >
> > Overall, I think it's an interesting paper and decide to raise my score.

---

> > > ### Author Response · Authors · 2021-11-30
> > > **Follow-up Response to Reviewer GSc9**
> > >
> > > Many thanks for your appreciation of our paper.
> > >
> > > **Q1-1.[MRKLD+LREND with center aware filtering]**
> > > It's a great advice. We spent the past day conducting the experiments and the results are listed as below (marked as bold). The higher performance of MRKLD+LRENT+Ca compared to MRKLD+LRENT implies the effectiveness of center-aware filtering. We also find that it achieves a lower performance than Tw+Ca. As MRKLD+LRENT itself contains a filtering process to select samples, a possible explanation is that MRKLD+LRENT may filters out too many samples which can be handled by our cross-attention mechanism. Although the precision of the pseudo labels in MRKLD+LRENT+Ca is 96.4% (higher than Tw+Ca 91.3%), the target recall of MRKLD+LRENT+Ca is only 20.0%, much lower than the recall of Tw+Ca 94.8%. It is worth mentioning that we started running the experiments immediately after reading your feedback, and there was not enough time for us to tune the parameters in MRKLD+LRENT+Ca, but used the default values instead.
> > >
> > >
> > > | **Pseudo labels** | **plane** | **bcycl** | **bus** | **car** | **horse** | **knife** | **mcycl** | **person** | **plant** | **sktbrd** | **train** | **truck** | **Avg.** |
> > > | --- | --- | --- | --- | --- | --- | --- | --- | --- | --- | --- | --- | --- | --- |
> > > | One-way-source | 96.1 | 52.7 | 85.5 | 69.6 | 95.0 | 90.2 | 95.1 | 66.6 | 88.8 | 54.6 | 95.4 | 29.5 | 76.6 |
> > > | One-way-target | 98.2 | 32.0 | 87.7 | 84.1 | 95.5 | 89.9 | 98.3 | 66.8 | 95.7 | 57.5 | 95.6 | 22.0 | 76.9 |
> > > | Two-way | 97.5 | 49.6 | 88.7 | 73.9 | 94.6 | 85.8 | 96.6 | 58.6 | 93.3 | 63.6 | 94.8 | 27.9 | 77.1 |
> > > | Tw+Ca | 98.1 | 86.9 | 87.9 | 80.9 | 97.9 | 97.3 | 96.8 | 85.3 | 97.6 | 83.2 | 94.0 | 54.4 | 88.4 |
> > > | RPLL | 98.4 | 63.4 | 85.8 | 68.8 | 97.0 | 95.4 | 97.8 | 59.3 | 96.2 | 57.2 | 96.2 | 48.1 | 80.3 |
> > > | MRKLD+LRENT | 97.8 | 77.3 | 81.4 | 64.3 | 94.6 | 93.9 | 93.3 | 77.5 | 93.1 | 74.9 | 92.6 | 59.0 | 83.3 |
> > > | **MRKLD+LRENT+Ca**| **97.5** | **74.6**| **85.3**| **75.7** | **95.4** | **96.6** | **94.7** |**73.7** | **93.6** | **78.7** | **90.7** | **49.4** | **83.8** |
> > > | Groundtruth | 97.9 | 89.1 | 92.3 | 91.9 | 98.4 | 97.2 | 97.5 | 86.8 | 98.6 | 90.7 | 96.3 | 60.0 | 91.5 |
> > >
> > >
> > > **Q1-2.[Our MRKLD+LRENT better than the original paper]**
> > > The original paper uses CBST as the base domain adaptation framework, whose backbone is ResNet101. In our re-implementation, MRKLD+LRENT is applied on the CDTrans framework with a backbone of DeiT for a fair comparison with our pseudo-labeling generation method. Therefore, the higher performance of our re-implemented MRKLD+LRENT results mainly come from the effectiveness of our CDTrans framework and the advantage of the transformer backbone.
> > >
> > > **Q2-1.[A simple self-attention module added to CNNs]**
> > > We agree that adding self-attention modules into CNNs could help improve the generalization ability of CNNs, like stated in [a]. But this variation does not necessarily outperform transformers. Transformers apply self-attention on all its layers besides using patches as inputs. Therefore, introducing self-attention into every layer in CNN might yield a much better result, maybe even better than transformers. However, the computational cost of doing so is unaffordable due to the large size of feature maps in CNN. That's why [a] only applies self-attention on the last block (c5) of Resnet50.
> > >
> > > **Q2-2.[Averaging useful? Increasing the conv kernel sizes?]**
> > > Although averaging could help to reduce the impact of noise, there are big differences between those in transformers and CNN. In CNN, the "averaging" is achieved by conv kernel and pooling layers, whose weights are fixed once the training is done. Besides, these averaging operations mainly have local scopes depending on the kernel size. In transformers, the "averaging" is achieved by self-attention. It is essentially a content adaptive kernel, the weights of which can be adaptively adjusted according to the content of input images. That's why the self-attention can distill noises from the patches. Moreover, self-attention has a global "averaging" scope on the whole image. Therefore, merely increasing the conv kernel size can only enlarge the "averaging" scope, but offers limited help for improving generalization ability due to the fixed weights.
> > >
> > > **Q3.[Robustness comes from patch?]**
> > > We agree with the reviewer that the patch-based approach plays a crucial role in transformers, which has also been proved in other works [b]. Meanwhile, we believe the self-attention mechanism in transformers is also very important for improving its robustness to noise. The strong generalization ability of transformers actually benefits from both factors.
> > >
> > > **Reference:**
> > >
> > > [a] "Bottleneck Transformers for Visual Recognition", CVPR2021
> > >
> > > [b] "Patches Are All You Need", ICLR2022 under review

---

> > > > ### Comment · Reviewer_GSc9 · 2021-12-01
> > > > **Thanks for the quick response**
> > > >
> > > > Thanks a lot for your quick responses. For MRKLD+LRENT, I think you can tune the thresholds to control the selected pseudo-label portion which could increase the recall for each class. But I agree it's hard tune the hyper-parameters in such a short time. Overall I think this is a good paper and will have impacts in domain adaptation.

---

> > ### Public Comment · ~Corey_Makowski1 · 2022-11-15
> > **Thanks.**
> >
> > Thanks for the brief info. I go to school. And I was looking for the top dissertation writing website. After that, I discovered the website at https://www.topwritersreview.com/reviews/buyessayfriend/, which has been a great help to me in finding the best essay writing service and also I can read reviews of those essay writing service before choosing them.

---

### Decision · Program_Chairs · 2022-01-20

**Decision:**

Accept (Poster)

**Comment:**

This paper makes use of cross-attention transformers to extract invariant features for unsupervised domain adaptation. Combined with pseudo-label approaches, the proposed method achieves state-of-the-art performance, possibly because the transformer features are more robust to the noise. In addition, a two-way centre-aware labeling method is proposed to produce more reliable pseudo labels.

However, there are some concerns raised by reviewers. After the discussion period, there is still a concern that is not completely unresolved. The comparison with existing methods might not be fair. It is possible that the performance gain is caused by the generally better representation of transformers, which has been shown in supervised image classification.

Overall, the paper is novel and interesting.  I would recommend acceptance of this paper given its impressive performance, but I highly suggest the authors add more ablation studies, for example, compare the proposed transformer and ResNet on a supervised classification task, to further confirm that the performance gain is solely because the transformer is more robust to label noise. The results can be updated in the supplementary. Also, as promised in the discussion, I hope the authors could release their code as soon as possible. This is because the backbone in this paper is totally new, it will be hard for other researchers to achieve SOTA results if they still use CNNs.